# Preliminary Data on Echocardiographic Evaluation and Serum Taurine Concentration in Healthy Dogs of Two Breeds (10 Golden Retrievers and 12 German Shorthaired Pointers) with Different Predispositions to Nutritional Dilated Cardiomyopathy: A Pilot Study

**DOI:** 10.3390/ani12212924

**Published:** 2022-10-25

**Authors:** Mara Bagardi, Sara Ghilardi, Iris Castellazzi, Eleonora Fusi, Michele Polli, Giulietta Minozzi, Stefano Faverzani, Caterina Mirabelli, Paola G. Brambilla

**Affiliations:** 1Department of Veterinary Medicine and Animal Sciences, University of Milan, Via dell’Università n. 6, 26900 Lodi, Italy; 2Anicura Clinica Veterinaria Orobica, Via Isonzo n. 2/e, Azzano San Paolo, 24052 Bergamo, Italy

**Keywords:** dilated cardiomyopathy, taurine, echocardiography, Golden Retriever, dog, nutritional dilated cardiomyopathy, pilot study

## Abstract

**Simple Summary:**

Concerns about a diet-associated form of dilated cardiomyopathy (DCM) have recently been highlighted following an increase in cases of DCM in dog breeds not known to have a genetic predisposition for this syndrome, such as Golden Retrievers. One hypothesis is that this form of DCM might be associated with a low concentration of taurine. Furthermore, in human medicine it has been demonstrated that a low vitamin D concentration may induce a thinning of cardiomyocytes and the onset of congestive heart failure. Because taurine is involved in the digestion and absorption of fat and liposoluble vitamins, including vitamin D, the aim of this pilot study was to compare the serum levels of taurine and vitamin D and to compare echocardiographic variables in healthy dogs of two breeds, Golden Retrievers (GRs) and German Shorthaired Pointers (GSPs), with different predispositions to nutritional DCM. The preliminary findings from this small pilot study showed that GRs may present lower levels of serum taurine and more altered echocardiographic variables than GSPs.

**Abstract:**

Dilated cardiomyopathy (DCM) is the most common myocardial disorder in dogs, and it is primarily considered to be an inherited or genetic disease with a higher prevalence in specific breeds such as Doberman Pinschers and Great Danes. Recently, several publications have reported concerns about cases of DCM in unusual breeds (Golden Retrievers—GRs) and associated them with specific diets (grain-free, high in pulses or potatoes, or low in taurine and amino acid precursors). Because taurine is involved in the digestion and absorption of fat and liposoluble vitamins, including vitamin D, the aims of this pilot study were: (1) to compare serum taurine and serum vitamin D (both implicated in cardiac function and absorbed from food) between healthy GRs and German Shorthaired Pointers (GSPs), breeds with different predispositions to nutritional DCM; (2) to highlight the differences between the echocardiographic variables in the two breeds; and (3) to evaluate the associations between the serum taurine and vitamin D concentrations and the echocardiographic features. Ten Golden Retrievers and twelve German Shorthaired Pointers were enrolled for complete hematobiochemical analyses, cardiac examinations, and serum taurine and vitamin D evaluations. The serum taurine concentrations were significantly lower in the GR dogs than in GSPs. All GRs were clinically healthy, but some echocardiographic variables, such as the sphericity index (related to left ventricle dilatation) as well as the end-systolic volume index and fractional shortening (both related to left ventricle systolic function), were different from the published reference ranges.

## 1. Introduction

Dilated cardiomyopathy (DCM) is a primary myocardial disease characterized by cardiac enlargement, eccentric hypertrophy, and the impaired systolic function of one or both ventricles. DCM is typically described as a hereditary or familial disease in breeds such as Dobermann Pinschers, Great Danes, and Irish Wolfhounds [1,2,3]. However, this disease can also be a consequence of chronic tachycardia, leading to cardiomyopathy, myocarditis, toxicity (e.g., doxorubicin), endocrine diseases (e.g., hypothyroidism), nutrient deficiency (e.g., taurine and carnitine), the end-stage form of other heart diseases (e.g., severe subaortic stenosis with afterload mismatch), post-resuscitation myocardial dysfunction, rare genetic/familiar disorders (e.g., endocardial fibroelastosis and Duchenne dystrophy), and systemic inflammatory response syndrome/sepsis [4,5,6,7,8,9,10,11,12,13,14,15]. Three stages of DCM can be characterized: the “occult” stage, during which the abnormalities are limited at the histological level and a careful cardiological examination may not diagnose it; the “preclinical” stage, during which the dog is asymptomatic, but a careful cardiological examination (echocardiogram and/or Holter ECG) can detect it; and the “overt” stage, where the clinical signs may include dyspnea, tachypnea, syncope, exercise intolerance, and cough [16]. During a physical examination, a soft systolic murmur may be auscultated at the left apex, and a tachyarrhythmia may be noted. The clinical signs, a physical exam, and chest radiography can help evaluate the patient, but echocardiography and Holter electrocardiograms (ECG) are central to the diagnosis of the preclinical and overt stages [4,6,16]. Finding more than 100 ventricular premature complexes (VPC) in a 24 h Holter ECG exam can suggest DCM in stages B1 (preclinical without systolic dysfunction or cardiac dilation) in breeds other than Dobermans and Boxers, in which the cut-off is 300 VPCs in a 24 h Holter ECG [16,17]. Echocardiographic findings predictive for DCM in Dobermans and other breeds are left ventricle (LV) systolic dysfunction and LV enlargement in diastole [1,16,17]. LV systolic dysfunction is characterized by increased systolic volumes (end-systolic volume—LVVs and end-systolic volume index—ESVI) and diameters (left ventricle internal diameter in systole—LVIDs and normalized left ventricular internal diameter in systole—LVIDNs), and reduced LV function parameters (fractional shortening—FS, left ventricular fractional area change—FAC, and ejection fraction—EF) [4]. LV enlargement is characterized by increased diastolic volumes (end-diastolic volume—LVVd and end-diastolic volume index—EDVI) and diameters (left ventricle internal diameter in diastole—LVIDd, normalized left ventricular internal diameter in diastole—LVIDNd, E-point to septal separation—EPSS, and left ventricle internal diameter in diastole to aortic root ratio—LVIDd/Ao), and increased sphericity of the chamber (sphericity index—SI) [11,12]. It is important to underline that some of these parameters, such as EPSS and SI, if considered alone, are not sufficiently sensible and specific as screening measurements in breeds such as Dobermans, unlike left ventricle volumes estimated using Simpson’s method of discs or Teicholz formula [1]. Nutritional DCM is a complex disorder with uncertain etiologies, an undetermined contribution of dietary factors, a variable phenotype, and unclear extra-dietary modifiers of disease expression [18,19]. An association between the development of DCM and low plasma and/or whole blood taurine has already been reported in American and English Cocker Spaniels [20]. Later on, different diets and specific ingredients were evaluated for possible associations with nutritional DCM: lamb and rice, diets high in fiber or low in protein, grain-free diets (the presence of potatoes or pulses in association with a particular source of animal protein, low in sulfur-containing amino acids, and/or low in taurine) [21,22,23,24,25,26]. In 2018–2019, the Food and Drug Administration (FDA) issued a warning and suggested that diets that are grain-free or contain legumes or potato ingredients need further studies to elucidate a possible role in the causation of DCM [27,28]. Of all breeds represented in the FDA report, the Golden Retriever (GR) is the most frequently reported breed to be affected by nutritionally mediated DCM. While there is no literature to support any familial relationship or genetic etiology in this breed, numerous data suggest that GRs may be more susceptible to taurine deficiencies and may need a higher dietary intake of this amino acid. These studies also demonstrated that dogs of this breed affected by dilated cardiomyopathy have a higher chance of partial/complete recovery when the diet is changed according to their needs and taurine is supplemented [29,30,31]. In the light of nutrients with cardioprotective effects, vitamin D must be mentioned. In the last years, more attention has been given to this nutrient because of its role in the regulation of bone metabolism and calcium homoeostasis. Cardiac myocytes express vitamin D receptors and a calcitriol-dependent calcium-binding protein. Low serum vitamin D concentrations were found to be correlated with a higher risk of congestive heart failure [18]. Moreover, it is well-known that in humans and companion animals taurine is involved in fat digestion and absorption. In particular, since taurine conjugates tend to remain in solution and are mostly absorbed in the ileum by an active mechanism, they are more available than glycoconjugates to form mixed micelles with fat. This would increase their intraluminal concentrations, with possible improvements in the absorption of fat and liposoluble vitamins, including vitamin D [26]. This has not yet been proven in dogs fed commercial diets. The aim of this pilot study was to compare serum taurine and vitamin D between healthy GRs and German Shorthaired Pointers (GSPs) (a breed similar to GRs in weight and body surface area (BSA) with no predisposition for DCM) to highlight the differences in echocardiographic variables between the two breeds and to evaluate the associations between serum taurine and the echocardiographic features of DCM.

## 2. Materials and Methods

### 2.1. Animals and Diet

This prospective cross-sectional pilot study included 22 privately owned dogs that visited the Cardiology Unit of the Veterinary Hospital of the Department of Veterinary Medicine, University of Milan, between March 2021 and December 2021. The informed consent was signed by the owners, according to ethical committee statement of the University of Milan number 2/2016. Groups were defined as GRs (n = 10; males = 3; females = 7) and GSPs (n = 12; males = 10; females = 2), aged from 1,5 to 7,5 years. All dogs were fed a complete, balanced diet. The inclusion criteria required that the dogs had an unchanged diet history for at least 3 months. All dogs were fed an adequate amount (considering age, weight, and the level of activity, according to FEDIAF nutritional guidelines) of food (divided into two meals). The diet was not included in the FDA’s reports and was not classified as associated with DCM [14,15]. The ingredients and chemical compositions of the diet are reported in Table 1. A medical history, a complete physical examination, hematobiochemical exams, a cardiological evaluation (electrocardiogram and a complete routine echocardiographic examination), and a nutritional evaluation (body weight, diet data, body condition score (BCS) and muscle condition score (MCS), according to WSAVA guidelines) were obtained for each dog enrolled in the study [32]. Furthermore, to estimate serum taurine and vitamin D, concentrated venous blood samples were collected from dogs after fasting (12–8 h).

The diet contained the following ingredients: salmon, corn, rice, dehydrated salmon protein, corn gluten meal, animal fats, corn semolina, soya flour, organoleptic viscera, dried beet pulp, dehydrated egg, minerals, dehydrated chicory root, fish oil, and soya oil.

### 2.2. Echocardiography

An echocardiographic examination (2-D, M-mode, spectral, and color-flow Doppler) was performed using a MyLab30 Gold cardiovascular ultrasound machine (Esaote, Genova, Italy) equipped with multifrequency phased array probes that were chosen according to the size of the subject. The exam was performed, conforming to a standard procedure without the requirement for sedation [33]. All measurements of interest were repeated by one single operator (MB), a third-year PhD student, with the supervision of an associate professor (PGB) with more than 25 years of experience in veterinary cardiology, on three consecutive cardiac cycles, and the mean value was used in the statistical analysis [34]. DCM was diagnosed only with both left ventricle dilatation and left ventricle systolic dysfunction [16,17]. The echocardiographic variables evaluated for the assessment of left ventricular size were: LVIDd and its value normalized for body weight using the allometric equation, as previously described (LVIDNd), EDVI, LVVd, EPSS, SI, and LVIDd/Ao [1,16,17,35,36,37]. The echocardiographic variables evaluated for the assessment of left ventricular systolic function were: LVIDs and its value normalized for body weight using the allometric equation, as previously described (LVIDNs), ESVI, LVVs, EF, FS, and FAC. The left ventricle internal diameter in diastole (LVIDs), FS, and FAC were obtained in M-mode, imaging from the right parasternal short-axis view at the level of the papillary muscle. The normalized left ventricle internal diameter in diastole and LVIDNs were considered normal if they were <1.65 and <1.1 cm/kg, respectively [17,35,36]. An FS > 25% and an FAC > 35% were considered normal [16,17]. The end-diastolic volume index and ESVI were calculated from LVIDd and LVIDs using the Teicholz method and were considered normal if EDVI < 100 mL/m^2^ and ESVI < 30 mL/m^2^ [35]. The left ventricle internal diameter in diastole was considered increased when >51 mm, and LVIDs was considered increased when >35 mm. These chosen cut-off values were based upon previously published data and breed-specific reference intervals [30,35,38]. The end-diastolic volume, LVVs, SI, and EF were obtained in B-mode, imaging from the left apical four chambers view. LVVd and LVVs were calculated using Simpson’s method of discs [16,17,39]. The sphericity index was considered normal if <1.65, and EF was considered normal if >40% [1,16,17]. The E-point to septal separation was obtained in M-mode, imaging from the right parasternal short-axis view at the level of the mitral valve leaflets and was considered normal if <7 mm [16,17]. The left ventricle internal diameter in diastole to aortic root ratio was obtained in B-mode, imaging from the right parasternal long-axis five chambers view and was considered normal if <2.6 [16,17]. The left ventricle internal diameter in diastole (LVIDs), LVVd, and LVVs were all evaluated based on the dog’s weight and according to reference intervals proposed in multibreed studies [1,16,17,35,36,39].

### 2.3. Hematobiochemical, Taurine, and Vitamin D3 Analyses

Venous blood samples were collected from all dogs to perform a standard blood analysis (complete blood count and biochemistry profile) and serum taurine and serum vitamin D3 (1,25-di-OH) concentration measurements. Serum taurine (0.5 mL) and vitamin D3 (1 mL) were analyzed in “Laboratorio di Analisi Veterinarie San Marco” (Veggiano, PD, Italy) and the IDEXX laboratory, respectively. The reference interval for serum taurine used in this study was 157 ± 52.6 nmol/mL, as reported in a multibreed study [40]. Thyroid profiles and troponin I concentrations were not evaluated in this study.

### 2.4. Statistical Analysis 

Statistical analyses were performed using SPSSTM 27.0 (IBM, Armonk, New York, NY, USA). Descriptive statistics were generated. The distribution of variables was assessed for normality with the Kolmogorov–Smirnov test. The variables are reported as means ± standard deviations. An unpaired *t*-test was used to compare the taurine and vitamin D values from the literature with the results of the study. Pearson correlations between the echocardiographic parameters and the serum taurine and vitamin D concentrations were considered weak, moderate, or strong, respectively, when the values of the correlation coefficients were less than 0.3, between 0.3 and 0.7, or more than 0.7. *p*-values < 0.05 were considered significant for all analyses. Probability values < 0.05 were considered significant.

## 3. Results

All analyzed variables were normally distributed. The complete blood counts and biochemical analyses were within the normal ranges for all dogs.

The included subjects were healthy at the complete cardiological examination. 

### 3.1. Golden Retrievers

Ten healthy GRs were included in the study (45% of the total population): seven (70%) females and three (30%) males. The median age was 3.30 ± 1.42 years (females 3 ± 1.15 and males 4 ± 2), and the median weight was 28.28 ± 3.54 kg (females 27.11 ± 3.49 and males 31 ± 2). All subjects had BCS values of 4–5/9 and normal MCS values. For both sexes, the weights were in line with the breed standard proposed by the American Kennel Club (AKC), which reports ideal weights between 29 and 34 kg for males and between 24 and 30 kg for females [41]. The serum taurine concentration was 108.94 ± 31.93 nmol/mL. This value is lower than the normal range (157 ± 52.6 nmol/mL) proposed in the literature for different breeds (*p* = 0.02) and lower than 178 nmol/mL (range, 110–272) [30,40]. There was no difference (*p* = 0.81) between the plasma taurine concentrations reported in the literature for GRs (106 ± 36.2 nmol/mL, ranging from 63 to 194 mmol/mL [31]) and the serum taurine concentrations obtained in the present study. The vitamin D3 (1.25-di-OH form) serum concentration was 284.88 ± 74.02 pmol/L. There were no significant correlations between the serum taurine and vitamin D3 concentrations and the echocardiographic parameters (*p* > 0.05) or between the serum taurine and vitamin D concentrations (*p* = 0.61). LVIDd and LVIDs were within the normal limits (43.37 ± 4.52 mm and 32.62 ± 3.93 mm, respectively), as was LVIDNd (1.63 ± 0.17 cm/kg) [17,35,36,37,38]. EDVI (92.23 ± 24.35 mL/m^2^) was within the limits, but its values were close to the maximum (100 mL/m^2^). ESVI was 47 ± 13.67 mL/m^2^, which was above the normal limit (30 mL/m^2^), as was LVIDNs (1.14 ± 0.14 cm/kg) [1,17,35,36]. LVVs and LVVd were normal (58.36 ± 9.46 mL and 24.06 ± 5.59 mL, respectively) [39]. The sphericity index was 1.49 ± 0.16 [1,16,17]. SF was 24.7 ± 5.58 %, which was lower than the normal value [16,17]. The LVIDd/Ao (2.02 ± 0.85), the EF (48.90 ± 9.46 %), the EPSS (0.54 ± 0.09 cm), the left ventricle FAC (40.63 ± 5.66 %), and the LA/Ao ratio (1.32 ± 0.15) were within the normal limits [16,17,41].

### 3.2. German Shorthaired Pointers 

Twelve healthy GSPs were included in the study (55% of the total population): two (16.6%) females and ten (83.3%) males. The median age was 4.58 ± 2.97 years (females 6.5 ± 4.95 and males 4.2 ± 2.66), and the median weight was 25.87 ± 3.22 kg (females 21.65 ± 0.49 and males 26.72 ± 2.81). All subjects had BCS values of 4–5/9 and normal MCS values. The serum taurine concentration was 164.28 ± 66.33 nmol/mL. This value is not statistically different from the normal serum taurine concentration (157 ± 52.6 nmol/mL) proposed in the literature (*p* = 0.77) [40]. The vitamin D3 (1.25-di-OH form) serum concentration was 238.25 ± 71.13 pmol/L. There were no significant correlations between the serum taurine and vitamin D3 concentrations and the echocardiographic parameters or between the serum taurine and vitamin D concentrations (*p* = 0.99). LVIDd and LVIDs were within the normal limits (42.44 ± 4.27 mm and 29.26 ± 2.94 mm, respectively) [35,37,38], as were LVIDNd and LVIDNs (1.63 ± 0.16 cm/kg and 1.05 ± 0.11 cm/kg, respectively) [17,35,36]. EDVI (92.60 ± 19.79 mL/m^2^) was within the limits, but its values were close to the maximum (100 mL/m^2^) [1,35]. ESVI was 38.02 ± 9.07 mL/m^2^, which was slightly above the normal limit of 30 mL/m^2^ [1,35]. LVVs and LVVd were normal (54.27 ± 9.86 mL and 26.65 ± 7.71 mL, respectively) [39]. The sphericity index was 1.48 ± 0.2. The LVIDd/Ao (2.18 ± 0.27), the EF (58.58 ± 5.32 %), the SF (30.75 ± 3.91 %), the EPSS (0.42 ± 0.10 cm), the left ventricle FAC (49.71 ± 4.78 %), and the LA/Ao ratio (1.14 ± 0.13) were within the normal limits [16,17,41]. 

### 3.3. Golden Retrievers vs. German Shorthaired Pointers 

The serum taurine concentration was higher in GSPs than in GRs (*p* = 0.021) (Table 2). In the overall population (considering both groups together), no significant correlations were observed between the serum taurine concentration (139 ± 59.47 nmol/mL) and age (4.00 ± 2.43 years) (*p* = 0.08) or weight (26.97 ± 3.51 kg) (*p* = 0.30). The vitamin D3 serum concentration was superimposable between GRs and GSPs (*p* = 0.16). No statistically significant correlations were observed between the vitamin D3 serum concentration and age (*p* = 0.49) or weight (*p* = 0.71). GRs showed greater LVID (*p* = 0.033), LA/Ao (*p* = 0.00), and EPSS (*p* = 0.008) and lower EF (*p* = 0.007), SF (*p* = 0.007), and left ventricular FAC (*p* = 0.002) compared to GSPs (Table 3).

## 4. Discussion

Nutritionally, DCM is a multifactorial disease whose etiopathogenetic origin has not yet been fully clarified. An increased sensitivity to taurine deficiency is reported in GRs, suggesting a key role of this amino acid in the development of DCM in this breed [29,30,31]. Our results appear to be in line overall with previous literature in light of the finding of a lower serum taurine concentration in the GRs, both compared to GSPs and to the serum normality ranges reported in the literature [30,40]. Although taurine deficiency is not the only cause of the development of DCM, this finding highlights the need to assess the real requirement for taurine (and/or its precursors) in the diet of GRs and, similarly, of other predisposed breeds (e.g., American Cocker Spaniel), as reported by Fascetti et al. in 2003 and Kaplan et al. in 2018 [22,30]. Moreover, the taurine concentration did not seem to influence the vitamin D concentration, as evidenced by the lower serum concentration present in GSPs. In dogs, the role of this amino acid in enhancing the absorption of fat-soluble vitamins was not found, as highlighted in human medicine [21], although further studies including more dogs are needed to conclusively confirm this.

All dogs considered in the present study were fed the same diet, which was not considered predisposing to the development of DCM [27]. 

Despite this, the serum taurine concentration was significantly lower in the GRs than in the GSPs. This highlights the need to conduct further studies both to clarify the actual role of diet in the bioavailability of taurine and to identify other causes related to the pathogenesis of the disease. The evaluation of serum taurine concentration is, to date, poorly documented in the literature, and there is only one study that took different breeds into consideration [40]. As for GRs, no specific serum ranges have been defined, while plasma and whole blood taurine levels have been reported [31]. There was no difference between our serum taurine concentration and the plasma taurine concentration reported by Ontiveros [31]. The authors are aware that different methods have been compared. The results suggest that the serum and plasma taurine concentrations may not differ, but further studies are needed to confirm this hypothesis. The small number of dogs included in the study does not allow us to propose these results as the serum taurine ranges for GRs. However, the obtained results can be considered a starting point for future studies for this purpose. Furthermore, all samples were processed in the same veterinary laboratory referral center (Clinica San Marco, Veggiano, PD, Italy). Therefore, analytical variability was reduced to a minimum. All GRs included in the study were clinically, hematobiochemically, and echocardiographically healthy. Despite this, some of the echocardiographic parameters, considered predictive of DCM in the literature, were above the upper limit reported in the multibreed studies or at least at the upper limit. Looking at the results of the published breed analysis in detail, it was found that (almost) all measurements of the non-sighthound breed GR were either within the generated body-weight-dependent prediction intervals or at least within the defined limits for “nondeviant breeds” (a breed was identified as a deviant breed if more than 10 % of the measurements of dogs of this breed were above or below the corresponding prediction intervals) [36]. The generally applicable prediction intervals can be used for this breed, although breed-specific M-mode values, if available, might be preferred. As reported in the literature, to evaluate left ventricular dilatation, EDV or LVVd, LVIDd and LVIDNd, and LVIDd/Ao and SI were considered [17]. In the GRs in this study, the sphericity index was lower than the value of 1.65 that was suggested as a cut-off by the 2003 canine DCM guidelines [37]. The same was observed for GSPs in the first hypothesis because of their hunting attitude and their high degree of physical training [43]. Therefore, although this finding is partly indicative, it is not possible to affirm whether it is a potential indicator of ventricular remodeling or is a normal finding for GRs. Unfortunately, the literature does not report breed-specific reference ranges for each echocardiographic variable that may be associated with systolic dysfunction and DCM development. Some studies have been published, but they mainly focused on the normalized left ventricular volumes or internal diameters and not on all the other variables considered in the present study. For this reason, the comparison with the published data was difficult and often not entirely correct (we often considered the reference ranges proposed for the Doberman, the most studied breed for this topic). To assess the systolic function of the left ventricle, ESV or LVVs, LVIDs and LVIDNs, EF, FS, left ventricular FAC, and EPSS [1,16,17,36,39] have been considered. ESVI was found to be above the upper normal limit (30 mL/m^2^) [1,35], but normal LVIDs, LVIDNs, and LVVs were obtained with Simpson method of discs [36,39]. Recent studies emphasize to combine the M-mode and B-mode measurements for an accurate evaluation of the echocardiographic parameters, allowing a correct diagnosis of DCM. This is even more important in those subjects for whom the presence of a geometric alteration of the heart chambers is suspected [17,44]. ESVI may overestimate the left ventricular volume and therefore may not be reliable. The FS was at the lower limit of the normal value of 25% [16,17]. Although this value alone does not allow the presence of systolic dysfunction to be defined, it is interesting to observe the low mean value in healthy GRs in our study. Furthermore, some echocardiographic parameters in the two breeds were statistically different despite their similar weights and BSAs. LVIDs, LA/Ao, and EPSS were significantly higher, whereas EF, FS, and left ventricular FAC were significantly lower in GRs than in GSPs. The higher values of the parameters suggestive of ventricular dilation and lower values of those predictive of systolic dysfunction, associated with low levels of serum taurine, support the hypothesis of a possible greater predisposition of GRs to develop DCM or simply suggest a peculiar breed characteristic that is not necessarily related to a lower plasma concentration of taurine. This may be confirmed by a study that compares the data of the two breeds and includes more than one hundred dogs of each breed that are fed an identical diet. At present, therefore, only assumptions can be made. No significant correlations were found between the echocardiographic parameters and the taurine and vitamin D serum concentrations. This result is probably due to the small number in the population under study but could also indicate other causal factors in determining the echocardiographic alterations. Furthermore, it must be highlighted that the literature does not report well-established echocardiographic normality ranges based on statistically reliable sample sizes for both GRs and GSPs. For this reason, it is difficult to correctly interpret the echocardiographic findings from this pilot study, which makes it easier to interpret as abnormal something that maybe could be normal for a given breed. Therefore, it is still necessary to clarify the role of vitamin D in the development of DCM and possibly congestive heart failure. To the authors’ knowledge, no veterinary study has published breed-specific data on this topic. In any case, further studies with larger populations are needed to better assess these correlations. The main limit of this study was the small number of subjects that were included, which inevitably weakened our statistical analysis and results. Accordingly, the findings from this small pilot study should not be read as a guide for the interpretation of the taurine concentration and echocardiographic parameters in these two breeds. In contrast, they should simply be interpreted as preliminary data aimed at paving the way for further and more robust investigations on this topic. Another important limitation of this study was the lack of follow-up of the included dogs. Therefore, a preclinical DCM in the included subjects cannot be ruled out. A further limitation was the determination of the serum taurine concentration. As discussed previously, this substrate is poorly described in the literature, and scarce information is available on its reliability and reference ranges compared to plasma. It is also worth noting that not all subjects were fed the same food, even though the analytical composition was the same. Although no diets indicated by the FDA report as predisposing to DCM were used, the obtained results would have been more homogeneous if all dogs were fed the same diet. 

## 5. Conclusions

Nutritionally dilated cardiomyopathy still needs in-depth studies before its complex etiological and pathogenetic mechanisms are completely clarified. In the present study, the serum taurine concentration was significantly lower in GR dogs than in GSPs and the reference ranges of plasma concentration reported for this breed and other breeds [21,31]. Unfortunately, the compositions of the diets in this study are not comparable with those reported in the literature, so the results are to be considered relative. All GRs were clinically healthy, but some echocardiographic variables such as SI (related to left ventricle dilatation) as well as ESVI and SF (both related to left ventricle systolic function) were different from the published reference ranges for DCM and were superimposable to the echocardiographic values reported for this breed [36,39]. To diagnose DCM, there must be systolic dysfunction and heart dilation, so these echocardiographic values, taken individually, cannot be considered predictive of DCM. For a non-equivocal evaluation of these subjects, it is desirable that the literature, in the future, reports breed-specific reference ranges for each echocardiographic variable that may be associated with systolic dysfunction, DCM development, and taurine deficiency. Although in this work it was not possible to correlate taurine deficiency with echocardiographic parameters, we assume that this hypothesis can be addressed and possibly confirmed by future prospective studies performed on larger populations. Similarly, the results obtained from this first evaluation of serum taurine concentrations can be seen as a starting point for future studies that will define the serum reference ranges for the GR breed.

## Figures and Tables

**Table 1 animals-12-02924-t001:** Chemical composition and ingredients of the diet.

**Chemical Composition**	**As Fed (%)**
Moisture	9.5
Protein	27
Fat	15
Carbohydrates	38
Fiber	3
Ash	7.5
Taurine (mg/kg)	750
Vitamin D3 (IU/kg)	1900
Calculated metabolizable energy (Kcal/kg) *	3550

* Metabolizable energy = 8.5 kcal ME/g fat + 3.5 kcal ME/g crude protein + 3.5 kcal ME/g nitrogen-free extract.

**Table 2 animals-12-02924-t002:** Taurine and vitamin D3 serum concentrations in the Golden Retrievers and German Shorthaired Pointers, their comparisons, and the reference ranges proposed in the literature.

	Overall Population	Golden Retrievers	German Shorthaired Pointers	*p*	Serum Reference Ranges	Plasma Reference Ranges
Taurine (nmol/mL)	139 ± 59.47	108.94 ± 31.93	164.28 ± 66.33	0.021 *	157 ± 52.6 (different breeds) [40] 178 (110–272) (different breeds) [30]	106 ± 36.2 (GR) [31] 77 ± 2.1 (different breeds) [21]
Vitamin D3 (pmol/L)	265.55 ± 73.27	284.88 ± 74.02	238.25 ± 71.13	0.16	Not published	Not published

* Statistically different; GR: Golden Retriever.

**Table 3 animals-12-02924-t003:** Echocardiographic variables in the Golden Retrievers and German Shorthaired Pointers, their comparisons, and the reference ranges proposed in the literature.

	Golden Retrievers	German Shorthaired Pointers	*p*	Reference Ranges
LVIDd (mm)	43.37 ± 4.52	42.44 ± 4.27	0.626	<51 [35,37,38]
LVIDs (mm)	32.62 ± 3.93	29.26 ± 2.94	0.033 *	<35 [35,37,38]
EDVI (mL/m^2^)	92.23 ± 24.35	92.60 ± 19.79	0.969	<100 [1,35]
ESVI (mL/m^2^)	47.00 ± 13.67	38.02 ± 9.07	0.080	<30 [1,35]
LVVd (mL)	58.36 ± 9.46	54.27 ± 9.86	0.084	Different references for non-sighthound and sighthound breeds [39]
LVVs (mL)	24.06 ± 5.59	26.65 ± 7.71	0.0763	
EF (%)	48.90 ± 9.46	58.58 ± 5.32	0.007 *	>40 [16,17]
SF (%)	24.70 ± 5.58	30.75 ± 3.91	0.007 *	>25 [16,17]
EPSS (cm)	0.54 ± 0.09	0.42 ± 0.09	0.008 *	<0.7 [16,17]
LA/Ao	1.32 ± 0.15	1.13 ± 0.13	0.005 *	1.05–1.73 [42]
SI	1.49 ± 0.16	1.48 ± 0.20	0.954	>1.65 [1,16,17]
FAC (%)	40.63 ± 5.66	49.71 ± 4.78	0.002 *	>35 [16,17]
LVIDd/Ao	2.3 ± 0.24	2.18 ± 0.27	0.341	<2.6 [16,17]
LVIDNd	1.63 ± 0.17	1.63 ± 0.16	0.931	1.17–1.63 (2.5–97.5 percentile) [17,35,36]
LVIDNs	1.14 ± 0.14	1.05 ± 0.11	0.108	0.7–1.09 (2.5–97.5 percentile) [17,35,36]

* Statistically different; LVIDd: left ventricular internal diameter in diastole; LVIDs: left ventricular internal diameter in systole; EDVI: end-diastolic volume index in diastole; ESVI: end-systolic volume index in systole; LVVd: left ventricle end-diastolic volume; LVVs: left ventricle end-systolic volume; EF: left ventricle ejection fraction; SF: left ventricle shortening fraction; EPSS: E-point to septal separation; LA/Ao: left atrium to aortic root ratio; SI: sphericity index; FAC: left ventricle fractional area change; LVIDd/Ao: left ventricle internal diameter in diastole to aortic root ratio; LVIDNd: left ventricular internal diameter in diastole; LVIDNs: left ventricular internal diameter in systole.

## Data Availability

The data presented in this study are all reported in the manuscript.

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
