# Peer review of "Preliminary Data on Echocardiographic Evaluation and Serum Taurine Concentration in Healthy Dogs of Two Breeds (10 Golden Retrievers and 12 German Shorthaired Pointers) with Different Predispositions to Nutritional Dilated Cardiomyopathy: A Pilot Study"

_animals, 2022, doi:10.3390/ani12212924_

Round 1
Author Response
The authors thank the Editor and the Reviewer for their thoroughly review of our study. We are very pleased about their appreciation of the work and their positive comments. Thank you very much.
We have carefully considered all reviewers’ comments and have tried to address them whenever we felt this was appropriate. We feel that the quality of our manuscript has improved following the Reviewers’ comments and suggestions.
Best regards
------------------------------------------------------------------------------------------------------------------------
Reviewer(s)' Comments to Author:
Reviewer: 1
The manuscript describes the echocardiographic comparison of two canine breeds and the analysis of the serum concentration of taurine and vitamin D. Although the study idea is interesting, the Authors did a great work trying to combing clinical/echocardiographic data with laboratory ones, and the manuscript is well-written, there are several weaknesses that significantly limit the scientific relevance of the manuscript in its current form, primarily the limited sample size. If this would have been the first research on this topic or on this breed, the current number of dogs, although limited, would have been acceptable (viewing it as a kind of pilot study). However, since many other studies have investigated this topic in this breed and these studies have included significantly large populations of dogs (e.g. “Development of plasma and whole blood taurine reference ranges and identification of dietary features associated with taurine deficiency and dilated cardiomyopathy in golden retrievers: A prospective, observational study” = 86 Golden; “Taurine deficiency and dilated cardiomyopathy in golden retrievers fed commercial diets” = 24 Golden with DCM and 52 healthy Golden; “Effect of type of diet on blood and plasma taurine concentrations, cardiac biomarkers, and echocardiograms in 4 dog breeds” = 37 Doberman Pinschers, 43 Golden Retrievers, 86 Whippets, and 22 Miniature Schnauzers), the current number of enrolled subjects made the interest related to this research and the reliability of the pertinent statistical analysis not strong enough to meet the standards of publication. According to the reviewer’s opinion, the study and control populations should increase at least of 3 times to strengthen the study enough to be published. Nevertheless, great attention has been paid to the revision of the current version of the manuscript, with the aim of providing useful suggestions to authors, so that they could work and improve accordingly their manuscript if they will try in future to increase the sample size of the study population and submit again their interesting research. Below, some concerns, questions and suggestions.
The Authors thank the Reviewer 1 for his/her thoroughly review, comments and suggestions. We really appreciate it.
We are aware of the limit of the samples number. However, this is the first study that evaluates the serum (not plasmatic) concentration of taurine in Golden Retrievers. Our Italian reference laboratory (Laboratorio Analisi Veterinarie San Marco – Veggiano - Padova) perform the analysis on this substrate and for our Department sending samples abroad was unmanageable for costs and fees. We have already performed the hematobiochemical analysis for the included subjects and the obtained values were within the reference ranges.
We have applied for a private award and have required funds for a project involving at least 40 Golden retrievers. We have already collected blood samples for the evaluation of the complete hematobiochemical examinations, urine analysis, serum taurine, troponin I and thyroid profile.
As soon as we have the economic availability it will be our intention to send samples for analysis and expand the statistics to obtain more reliable data.
For the moment, as suggested, we have stated that this is a pilot study.
Simple Summary
-In the Simple Summary it has been explained why Authors wanted to study taurine concentration in Golden; however, it is clear neither why they also wanted to measure vitamin D nor why they specifically selected German Shorthaired Pointers. These points should be clearly explained to better introduce the study background.
Vitamin D has several cardioprotective functions including inhibition of the renin-angiotensin-aldosterone system and of proinflammatory cytokines and reduces the cardiomyocyte hypertrophy (Patel and Rizvi, 2011). In human medicine, vitamin D deficiency has long been associated with the development of congestive heart failure and its integration is an important factor in the prevention of this condition (Wang et al., 2008). As for the dog, there is only one study in the literature in which it has been observed that reduced concentrations of serum vitamin D are associated with a higher risk of heart failure and a worse prognosis (Kraus et al., 2014). For these reasons it also seemed interesting to evaluate the serum concentration of this vitamin in the included subjects and to correlate it with the concentration of taurine, essential amino acid for a function.
In fact, it is well known that in humans and companion animals, taurine is involved in fat digestion and absorption. In particular, since taurine conjugates tend to remain in solution and are mostly absorbed in the ileum by an active mechanism, they are more available than glycoconjugates to form mixed micelles with fat (1). This would increase their intraluminal concentrations with possible improvement in absorption of fat and liposoluble vitamins.
The evaluation of echocardiographic parameters indicative of DCM and its association with serum concentration of taurine and vitamin D in a predisposed breed (Golden Retriever) and not (German Pointers) could shed light on the characteristics and etiopathogenetic aspects of this disease and the breeds’ differences.
-“The overall aim was to evaluate differences between the two breeds” This sound too generic. Please rewrite to specify which types of differences you were looking for.
Thank you very much, we have modified the sentence as follow: “The aim of this pilot study was to compare serum levels of taurine and vitamin D, and perform and compare echocardiographic evaluation variables in two healthy dog breeds, Golden Retrievers and German Shorthaired Pointers”.
-“These findings suggest that Golden Retrievers may be more subject to DCM than other breeds due to their taurine deficiency”. This sentence should be rewritten as you did not compare Golden with other breeds, but only with German Shorthaired Pointers.
Thank you very much for this comment, we have decided to delete this sentence.
Abstract
-Also here, as in said before for the Simple Summary, it should be explained why you decided to measure also vitamin D and why you selected German Shorthaired Pointers as control group.
Thank you for this suggestion. We decided to modify the sentence as follow: “The aims of this study were three: 1) compare serum taurine and serum vitamin D (both implicated in cardiac function and carried out with food) between healthy Golden Retrievers and German Shorthaired Pointers”.
-“Although all Golden Retrievers were healthy, some echocardiographic measures predictive for DCM were abnormal: sphericity index, related to left ventricle dilatation, end-systolic volume index and fractional shortening, both related to left ventricle systolic function.” This sentence needs grammar revision and should be rephrased to sound clearer.
Thank you very much for this comment. We have rephrased the sentence, as suggested: “All Golden Retrievers were clinically healthy, but some echocardiographic variables such as sphericity index (related to left ventricle dilatation), and end-systolic volume index and fractional shortening (both related to left ventricle systolic function) were different from the reference ranges.”
Introduction
-lines 48-49: the list of differentials for a DCM-like phenotype is incomplete. Additional DD include for example end-stage form of other heart disease (e.g., severe SAS with afterload mismatch), postresuscitation myocardial dysfunction, rare genetic/familiar disorders (e.g., endocardial fibroelastosis, Duchenne dystrophy), SIRS/sepsis. Useful references are: “Endocardial fibroelastosis in a dog with congestive heart failure”; “Reversible myocardial dysfunction in a dog after resuscitation from cardiopulmonary arrest”; “Reversible myocardial depression associated with sepsis in a dog”; “Natural History of Cardiomyopathy in Adult Dogs with Golden Retriever Muscular Dystrophy”. Moreover, the terms chronic tachycardia, toxicity, endocrine diseases, and nutrient deficiency are too generic and should be contextualized, for example saying: “chronic tachycardia leading to tachycardiomyopathy, toxicity (e.g. doxorubicin), endocrine diseases (e.g. hypothyroidism), and nutrient deficiency (e.g., taurine, carnitine)”.
Thank you very much for these suggestions. We have added the related references and modified the text.
-Lines 49-52: actually, the stages of DCM are three, not two: 1) the “occult” one, during which the abnormalities are limited at the histological level and also a careful cardiological examination may not diagnose it, 2) the “preclinical” one, during which the dog is asymptomatic but a careful examination can detect it, and 3) the “overt” one.
Thank you very much for this comment. We have modified the text and have added the correct recent reference.
-Lines 56-57: “Finding more than 100 ventricular premature complexes (VPC) in a 24h-Holter ECG exam can suggest preclinical DCM in suspected cases [4]”. Although I personally agree with such a cut-off, it is the traditional/old one and a more recent one with different criteria has been proposed by Prof. G. Wess. Please provide also it to readers with proper reference.
Thank you, we have modified the sentence and added the proper reference (Wess 2022).
-From line 57 to line 67 Author list a large number of echocardiographic variables using reference n. 4. However, this reference is a state of the art/review; if we take a look to the original studies on echo of Dobby with DCM we can see that some echocardiographic variables are less important than others. Therefore, I suggest to Authors: 1) or to list only the two best predictors of DCM in Dobby (EDVI and ESVI > 95 and 55 by Simpson method, respectively), ore 2) to maintain all echo variables but specifying which are the most important for diagnosis of DCM in Dobby (e.g., sphericity index and EPSS have not the same diagnostic value of EDVI and ESVI) by using proper references.
Thank you very much for this comment. We agree with the Reviewer, different parameters have different diagnostic significance for DCM in Dobby. However, we would not focalize the introduction on Dobby and so we decided to not list only the two best predictors of DCM in this breed and so to maintain all echo variables but specifying that SI and EPSS are the less important. We have modified the text.
-Additional consideration: sincerely, I do not see the need for a section about the diagnostic criteria of primary DCM from Dobby literature if the topic of the manuscript is DCM secondary to taurine deficiency in Golden. It seems that the introduction of the manuscript should be modified, removing several sentences to jumping directly to the real topic of the study.
Thank you very much for this comment and consideration. We decided to modified this information trying to be more direct to the focus of the manuscript. Thank you.
M&M
-The criteria used to determine if a dog was healthy or not seems not complete. Indeed, in a similar study, also routine blood works (CBC, biochemistry) and thyroid profile seems useful to be sure that systemic disorders (including hypothyroidism) able to mimic a DCM-like phenotype are not present. This section should be made more robust to allow this study to be published.
Thank you very much for this comment. Yes, as stated before, we have performed the hematobiochemical evaluation of all included subjects. We have not performed the thyroid profile with T4 and TSH because none of the subjects presented CBC and biochemical alterations that can be correlated with hypothyroidism (microcytic and hypochromic anaemia and increase in cholesterol). We are aware of the need to perform this evaluation in the included dogs, as well as the evaluation of plasmatic troponin. Hypothyroidism may influence food intake, body weight and muscle condition, but this variable was not evaluated in this study. Hypothyroidism has also long been considered a variable that may contribute to reduced systolic function (Guglielmini C, Berlanda M, Fracassi F, Poser H, Koren S, Baron Toaldo M. Electrocardiographic and echocardiographic evaluation in dogs with hypothyroidism before and after levothyroxine supplementation: A prospective controlled study. J Vet Intern Med. Wiley Online Library; 2019). However, recent studies confirm that the key variables of systolic function and chamber dilation measured in this study (FS, LVIDd and LVIDs) were not significantly different between dogs with untreated hypothyroidism and control dogs. Significant hypothyroidism is not expected in any of the study participants, as they were free of clinical signs of this condition. Additionally, more rigorous screening tests were not performed as they were determined to be beyond the scope and budget of this study. While the yield of these diagnostic tests in patients without clinical signs is expected to be low, future studies may consider cardiac biomarkers (troponin) as part of the health screening at time of enrolment. These additional screening tests may be helpful in ruling out rare conditions that may contribute to reduced systolic function. As declared, we have applied for a private award and have required funds for a project involving at least 40 Golden retrievers. We have already collected blood samples of other 35 Golden Retrievers for the evaluation of the complete hematobiochemical examinations, urine analysis, serum taurine, troponin I and thyroid profile. As soon as we have the economic availability it will be our intention to send samples for analysis and expand the statistics to obtain more reliable data.
-Line 105: Were diets standardized? Were these diets commercially available or home-made? How many dogs received a commercial diet and how many a home-made one? Was there a standardization in term of timing between the last feeding and the blood sample acquisition? Please, provide more data at regard as this is the main topic of the study and the pertinent methodology should be rigorous and well described.
Thank you, yes, all the included dogs received the commercially dry diets with the same composition. All dogs had been fasting for at least 12 hours before the blood sampling (even if fasting status does not impact taurine status in dogs - Gray K, Alexander LG, Staunton R, Colyer A, Watson A, Fascetti AJ. The effect of 48-hour fasting on taurine status in healthy adult dogs. J Anim Physiol Anim Nutr (Berl). Germany; 2016; 100: 532–536. https://doi.org/10.1111/jpn.12378 PMID: 26250395). We have modified the text and added thise information in order to be more pertinent in the description of the methodology.
-Echocardiographic cut-off: authors used widely-accepted cut-off. I wonder if it would be more appropriate for Golden Retriever to use breed-specific cut-off. Some data on echocardiographic variables in this breed can be found in literature. Since this breed may have specific breed-related features, I suggest to author to review their echocardiographic measurements to check if their final diagnosis would remain the same by using the breed-specific values or not (I think that some dogs judged to be as DCM-positive would be reclassified as normal).
Thank you very much for this suggestion. We have added the proper literature references and compared our results with these data, modifying the discussion. In particular: The fractional shortening was recorded as low when <25%; LVIDd was considered increased when >51 mm, and LVIDs was considered increased when >35 mm. These chosen cut-off values are based upon previously published data and breed specific reference intervals which also maintain consistency with prior publication (Kaplan JL, Stern JA, Fascetti AJ, Larsen JA, Skolnik H, Peddle GD, et al. Taurine deficiency and dilated cardiomyopathy in golden retrievers fed commercial diets. PLoS One. United States; 2018; 13: e0209112; Morrison SA, Moise NS, Scarlett J, Mohammed H, Yeager AE. Effect of breed and body weight on echocardiographic values in four breeds of dogs of differing somatotype. J Vet Intern Med. United States; 1992; 6: 220–224; Cornell CC, Kittleson MD, Della Torre P, Haggstrom J, Lombard CW, Pedersen HD, et al. Allometric scaling of M-mode cardiac measurements in normal adult dogs. J Vet Intern Med. United States; 2004; 18: 311–321).
-Lines 153-154: “The reference interval for serum taurine used in this study was considered 157 ± 52,6 nmol/ml as reported in a multi-breed study [25]” Why authors did not used whole blood or plasma, two hematologic sources for which criteria have been established in Golden (“Development of plasma and whole blood taurine reference ranges and identification of dietary features associated with taurine deficiency and dilated cardiomyopathy in golden retrievers: A prospective, observational study”)?.
Thank you very much for the comment. We hope to have already answered the Reviewer about this issue in the first paragraph of this letter. Basically, there were reasons related to the choice of the laboratory that performed the analysis.
Moreover, if we look at the reference range of plasma taurine in Golden from the previously cited manuscript (i.e., 63 to 194 nmol/mL), it is in line with results from this study (i.e., s 108.94±31.93 nmol/ml). By this view, the values detected by the authors could be seen as normal. At this point, a reply to my observation may be that is not completely correct to compare a value obtained by serum with another one obtained by plasma (and I could agree with this statement). Nevertheless, the authors did such a type of comparison in their study (lines 179-180). Therefore, more clarification on the rationale behind the specific choices of the authors should be made (both their choice of using serum instead of whole blood or plasma, and their choice concerning reference of comparison).
Thank you very much for the comment. This topic is much felt by our research group because we realized that there are no serum ranges for taurine in different breeds fed with commercial diets. Normal reference ranges for whole blood and plasma taurine can be established based on a small handful of studies published in 2003. In a study by Torres et al, mean +/- standard deviation of plasma and whole blood taurine concentrations in 12 healthy beagle dogs fed purified diets were 109 +/-8 nmol/ml and 291 +/-25 nmol/ml, respectively (Meurs KM, Lahmers S, Keene BW, White SN, Oyama MA, Mauceli E, et al. A splice site mutation in a gene encoding for PDK4, a mitochondrial protein, is associated with the development of dilated cardiomyopathy in the Doberman pinscher. Hum Genet. Germany; 2012; 131: 1319–1325.). A larger study by Delaney et al reported mean +/- standard error of plasma and whole blood taurine concentrations in 131 apparently healthy dogs of a variety of breeds consuming commercial diets to be 77 +/-2.1 nmol/ml and 266 +/-5.1 nmol/ml, respectively (Simpson S, Dunning MD, Brownlie S, Patel J, Godden M, Cobb M, et al. Multiple Genetic Associations with Irish Wolfhound Dilated Cardiomyopathy. Biomed Res Int. United States; 2016; 2016: 6374082.). Serum taurine concentrations, although less repeatable, in apparently healthy dogs were recently reported to have a median of 178 (range, 110–272) (Werner P, Raducha MG, Prociuk U, Sleeper MM, Van Winkle TJ, Henthorn PS. A novel locus for dilated cardiomyopathy maps to canine chromosome 8. Genomics. United States; 2008; 91: 517–521). Unfortunately, this study did not report the composition of the food administered to the animals and no study compared serum and plasma taurine concentration in dogs with standardized diets. Furthermore, there is little understanding of how serum, plasma and whole blood taurine concentrations correlate with myocardial taurine concentrations in the dog, as endomyocardial biopsy is challenging and not clinically practical. To confound the assessment of taurine concentrations even further, the degree to which breed- or size-specific reference ranges are necessary is unknown but is supported by data that establish specific breeds like Golden Retrievers, Newfoundlands and American cocker spaniels as having a greater risk for taurine deficiency and subsequent cardiac disease (Freeman LM, Rush JE. Nutrition and cardiomyopathy: lessons from spontaneous animal models. Curr Heart Fail Rep. United States; 2007; 4: 84–90. Yost O, Friedenberg SG, Jesty SA, Olby NJ, Meurs KM. The R9H phospholamban mutation is associated with highly penetrant dilated cardiomyopathy and sudden death in a spontaneous canine model. Gene. Netherlands; 2019; 697: 118–122. Harmon MW, Leach SB, Lamb KE. Dilated Cardiomyopathy in Standard Schnauzers: Retrospective Study of 15 Cases. J Am Anim Hosp Assoc. United States; 2017; 53: 38–44. Martin MWS, Stafford Johnson MJ, Celona B. Canine dilated cardiomyopathy: a retrospective study of signalment, presentation and clinical findings in 369 cases. J Small Anim Pract. England; 2009; 50: 23– 29. Wiersma AC, Stabej P, Leegwater PAJ, Van Oost BA, Ollier WE, Dukes-McEwan J. Evaluation of 15 candidate genes for dilated cardiomyopathy in the Newfoundland dog. J Hered. United States; 2008; 99: 73–80. Meurs KM, Stern JA, Sisson DD, Kittleson MD, Cunningham SM, Ames MK, et al. Association of dilated cardiomyopathy with the striatin mutation genotype in boxer dogs. J Vet Intern Med. United States; 2013; 27: 1437–1440). In our pilot study, serum taurine concentration was 108.94±31.93 nmol/ml in Golden Retrievers fed with standardized commercially dry food. This value is lower than the normal range (157±52.6 nmol/ml) proposed by literature for different breeds (p=0.02) but within the range of 63-194 nmol/ml proposed for plasma by Ontiveros et al. in 2020. However, the dogs were not fed with the same commercial diet, but it was declared that this range was for dogs fed with “traditional” diets (so not grain free). According to the unpaired t-test (that little improperly, compared the values from literature with the results obtained in the present study) there is no difference (p=0.81) between the plasma taurine concentration reported by literature for Golden Retrievers (106±36.2 nmol/ml) and the serum taurine concentration obtained in the present study. Although these comparisons are all a little bit improper, this is an interesting result because for the first time in veterinary medicine a serum range for this canine breed has been proposed. Obviously, a greater cohort of subject is needed to validate the data. We hope to have clarified our intentions. Thank you very much for the comment and for the opportunity to discuss about this.
-Evaluation of echocardiographic variable in Golden and comparison of echocardiographic variable with controls: I’m not completely sure that the values observed are really diagnostic for DCM in all Golden. The problem may arise by the selection of generic cut-off. Indeed, it has been demonstrated that it is not unusual for Golden to have one or more measurements, including those related to LV size, above the generic cut-off, sometimes also significantly (i.e., >10%) (“Left ventricular M-mode prediction intervals in 7651 dogs: Population-wide and selected breed-specific values”: in this study this has been observed by studying 89 Golden). Another source of concerns is related to the limited number of dogs, who make impossible to obtain statistically robust results as well statistically reliable reference ranges to be compared between breeds. Lastly, it cannot be conclusively demonstrated that the differences observed are simply the result of physiologically different cut-off between two distinct breeds rather than a pathologic effect of a true taurine deficiency in one of the two breeds. A larger study population would allow to eliminate some doubts at regard.
Thank you, we completely agree with Reviewer 1 and we have modified the references for the echocardiographic measurements for Golden Retrievers as previously suggested. Regarding the different cut-off between two distinct breeds rather than a pathologic effect of a true taurine deficiency the Reviewer’s claim should be confirmed by a larger ventricular volume of the Golden Retrievers. This is not proved by literature (Echocardiographic reference intervals for volumetric measurements of the left ventricle using the Simpson's method of discs in 1331 dogs), that shows normally bigger left ventricle SMOD volumes in sighthound breeds. This is not observed in our results.
-Comparison of taurine levels between Golden and controls: also in this case, it has not been conclusively demonstrated if the differences are related to a true, pathological deficiency or if they are simply the results of physiologically different concentration of a blood value between these two breeds. Again, also this may be better clarified by increasing the study and control population (and also by increasing the number of tests aimed at demonstrating that enrolled dogs are really healthy, and by standardizing the diet).
Thank you for this comment. We have declared that dogs were fed with commercially dry foods with the same composition. However, we completely agree that the results may be better clarified by increasing the study and control population (and also by increasing the number of blood tests aimed at demonstrating that enrolled dogs are really healthy, adding also thyroid profile and troponin).
-Correlation between taurine levels and levels of vitamin D3: author performed such a type of statistical analysis; however, it has not been explained to readers the rational behind this analysis. Does it exist some pathophysiological link between taurine and vitamin D3? Please provide more data at regard in the introduction and/or discussion.
Thank you for the comment. We have previously tried to respond at this query and modified the text.
Discussion
The discussion is well written and correctly analyzes the different key point of the study. However, in several parts of the Discussion, Authors have explained that their results were biased by the limitations of the studies (primarily the limited sample size, the type of hematologic source used for taurine concentration and the cut-off used for echocardiography). As almost all results are affected by these biases, it seems that Authors agree with my concerns. I sincerely believe that, as the study idea is interesting, this research should not be put apart by Authors but, in contrast, it merits to be strengthened and expanded, primarily by increasing the sample size at least of 3 times and providing more information on enrolled dogs. At the time, I strongly encourage Authors to resubmit the manuscript, as it will be more robust and mature for publication.
We are very pleased that Reviewer 1 appreciates the theme of our work and believes that the population should be expanded. As stated, we are working in this direction, and we have already collected a lot of samples and echocardiographic data. We are unfortunately waiting for the funds for taurine, troponin, and thyroid profiles, and we did not want to miss the opportunity to publish the preliminary data of this project reported in this manuscript. Therefore, we have tried to respond to all the requests of all Reviewers, trying to make the revision as complete as possible, hoping to have solved at least the major critical points. Unfortunately, the number of dogs included remains the biggest limit, and for the moment we cannot remedy this. We rely on the decision of reviewers.
Thank you so much

Reviewer 2 Report
Dear Authors,
The study is interesting because it provides data on the serum taurine concentrations in two breeds of dogs. However, you should consider the following recommendations previous to publication:
Title:
Line 2. The tithe should be modified since no dog in the study had dilated cardiomyopathy. Therefore, it could be as follows: “Echocardiographic evaluation and serum taurine concentration in two healthy dog breeds, Golden Retrievers and German Pointers, with different predisposition to nutritional dilated cardiomyopathy
Small abstract:
Line 19. Please include “… healthy two dog breeds….”
Line 21 and 22: please remove “The overall aim was to evaluate differences between the two breeds” because it is explained previously.
Abstract:
Line 33. Please indicate that both groups of dogs are healthy.
Line 36: Please rephrase this sentence, including only “the echocardiographic measurements xxxx are different from the reference ranges.
Introduction
Aim of the study: Please indicate that dogs are healthy
Line 171. Please substitute cases by Golden retriever dogs. Also, in tables
Line 195. Please substitute controls by German Pointers. Also, in tables.
Materials and methods
Line 103: Please indicate groups and remove the word “cases” because all dogs are healthy
Line 106: Do the ingredients correspond to a commercial diet? Please indicate
Line 125: Is Dr. MB a board certificate in cardiology? Please indicate qualification
Results
Please remove tables 1 and 2 because the echocardiographic values are included in the text, and also in table 3.
Line 225. Table 3. Echocardiographic measurements of the two breed dogs.
In the table 3, please substitute “cases” by Golden Retrievers and “control” by German Pointers. Please, include reference ranges also. In the bottom of the table 3 indicate the meaning of the abbreviations (LVIDd …..).
Please, results of taurine should be included in a table as well as the reference ranges.
Discussion:
Line 255-256: “In the group of Golden Retrievers of this study, the sphericity index was lower than the value of 1.65 suggested as a cut-off by the 2003 canine DCM guidelines”. Please explains why German Pointers have also lower SI values than reference ranges.
Line 283: The main limit of this study was the small number of animals included.
Another important limitation of the study is that the dogs have not been followed clinically and therefore it cannot be ruled out that they are in a preclinical phase.
Conclusions:
All conclusions are not derived from the present study so they must be rewritten. You must include and interpret the main findings of the study
Author Response
The authors thank the Editor and the Reviewer for their thoroughly review of our study. We are very pleased about their appreciation of the work and their positive comments. Thank you very much.
We have carefully considered all reviewers’ comments and have tried to address them whenever we felt this was appropriate. We feel that the quality of our manuscript has improved following the Reviewers’ comments and suggestions.
Best regards
------------------------------------------------------------------------------------------------------------------------
Reviewer(s)' Comments to Author:
Reviewer: 2
Dear Authors,
The study is interesting because it provides data on the serum taurine concentrations in two breeds of dogs. However, you should consider the following recommendations previous to publication:
Thank you very much for your appreciation of the work, we are really pleased. We have tried to follow all your recommendations and suggestions. Thank you very much for your revision.
Title:
Line 2. The tithe should be modified since no dog in the study had dilated cardiomyopathy. Therefore, it could be as follows: “Echocardiographic evaluation and serum taurine concentration in two healthy dog breeds, Golden Retrievers and German Pointers, with different predisposition to nutritional dilated cardiomyopathy
Thank you very much for your suggestion. We agree with the Reviewer, and we have modified the title.
Small abstract:
Line 19. Please include “… healthy two dog breeds….”
Line 21 and 22: please remove “The overall aim was to evaluate differences between the two breeds” because it is explained previously.
Thank you, we have modified as suggested.
Abstract:
Line 33. Please indicate that both groups of dogs are healthy.
Line 36: Please rephrase this sentence, including only “the echocardiographic measurements xxxx are different from the reference ranges.
Thank you very much, we have modified as suggested.
Introduction
Aim of the study: Please indicate that dogs are healthy
Line 171. Please substitute cases by Golden retriever dogs. Also, in tables
Line 195. Please substitute controls by German Pointers. Also, in tables.
Thank you very much for this suggestion. We have modified as suggested.
Materials and methods
Line 103: Please indicate groups and remove the word “cases” because all dogs are healthy
Thank you for the comment, we have modified the text and we have not used the terms “cases” and “controls”.
Line 106: Do the ingredients correspond to a commercial diet? Please indicate
Thank you for the comment. We have modified this paragraph in order to be more precise in the description of the diets.
Line 125: Is Dr. MB a board certificate in cardiology? Please indicate qualification
Thank you, we have modified as suggested and we have added the required information.
Results
Please remove tables 1 and 2 because the echocardiographic values are included in the text, and also in table 3.
Line 225. Table 3. Echocardiographic measurements of the two breed dogs.
In the table 3, please substitute “cases” by Golden Retrievers and “control” by German Pointers. Please, include reference ranges also. In the bottom of the table 3 indicate the meaning of the abbreviations (LVIDd …..).
Please, results of taurine should be included in a table as well as the reference ranges.
Thank you very much for all these suggestions. We have deleted tables n. 1 and 2 and added the reference ranges in the new table 2. We have also created a table 1 with taurine and vitamin D3 values and reference ranges for each group as required. Thanks to Reviewer 2 suggestions, the tables are now more complete. Thank you very much.
Discussion:
Line 255-256: “In the group of Golden Retrievers of this study, the sphericity index was lower than the value of 1.65 suggested as a cut-off by the 2003 canine DCM guidelines”. Please explains why German Pointers have also lower SI values than reference ranges.
Line 283: The main limit of this study was the small number of animals included.
Another important limitation of the study is that the dogs have not been followed clinically and therefore it cannot be ruled out that they are in a preclinical phase.
Thank you very much for these interesting comments. We have added the suggested speculations.
Conclusions:
All conclusions are not derived from the present study so they must be rewritten. You must include and interpret the main findings of the study
Thank you for the comment. We have modified the conclusions. Thanks to Reviewer 2 comments the conclusion in now more focused on the study. Thank you very much.
------------------------------------------------------------------------------------------------------------------------

Reviewer 3 Report
Dear Authors,
I found your paper interesting and new., I have only few remarks as follows:
line 31 : please use "variables" rather than "measures", all along the manuscript and tables.
line 129 : "as previously described" : I guess this sentence lacks a reference.
line 136 : "cm/kg" : please check the appropriateness
line: 137 : I encourage to calculate ESVI and EDVI in long axis views, especially as you used these views for measuring LVVs and LVVd
Line 204 : (p>0.05) please avoid
Line 246: please add "dogs"
Author Response
The authors thank the Editor and the Reviewer for their thoroughly review of our study. We are very pleased about their appreciation of the work and their positive comments. Thank you very much.
We have carefully considered all reviewers’ comments and have tried to address them whenever we felt this was appropriate. We feel that the quality of our manuscript has improved following the Reviewers’ comments and suggestions.
Best regards
------------------------------------------------------------------------------------------------------------------------
Reviewer(s)' Comments to Author:
Reviewer: 3
Dear Authors,
I found your paper interesting and new, I have only few remarks as follows:
Thank you very much for your appreciation of the work, we are really pleased.
line 31 : please use "variables" rather than "measures", all along the manuscript and tables.
Thank you, we have changed the term in all the manuscript when appropriate as suggested.
line 129 : "as previously described" : I guess this sentence lacks a reference.
Thank you for this suggestion, we have added the references and change the reference list.
line 136 : "cm/kg" : please check the appropriateness
Thank you for this comment. We have checked the reference paper (Cornell CC, Kittleson MD, Della Torre P, Häggström J, Lombard CW, Pedersen HD, Vollmar A, Wey A. Allometric scaling of M-mode cardiac measurements in normal adult dogs. J Vet Intern Med. 2004 May-Jun;18(3):311-21) and the unit in which this variable is expressed is correct, so we decided to not delete this reference unit. Thank you.
line: 137 : I encourage to calculate ESVI and EDVI in long axis views, especially as you used these views for measuring LVVs and LVVd
Thank you very much for this comment and suggestion. This is a very interesting topic. Wess has always described the obtainment of left ventricular volumes using right parasternal short axis view in Doberman with dilated cardiomyopathy. However, a recently published, large study of SMOD derived left ventricle volumes in more than 1300 dogs of varying breeds by Wess et al. [Echocardiographic reference intervals for volumetric measurements of the left ventricle using the Simpson’s method of discs in 1331 dogs. J Vet Intern Med 2021;35:724e38.] showed minimal bias (<1 mL) and good limits of agreement (about 6 mL) between the right parasternal long axis four chamber view and left apical four chamber view approaches for measuring EDV and ESV. This suggests that a single plane SMOD approach is reasonable if the image quality is good. In the present study we have always obtained both views, and area lengths are superimposable. We have added this reference.
Line 204 : (p>0.05) please avoid
Thank you, we have deleted this data.
Line 246: please add "dogs"
Thank you, added.
------------------------------------------------------------------------------------------------------------------------

Round 2
Reviewer 1 Report
The comments for Editors/Authors are present in the attached file named "Revision_1"

Author Response
I really thank the Authors for the work made during the revision. The manuscript appears improved thanks to these changes. I still have some doubts about the utility of publishing preliminary data from a small population of animals, because these data may be statistically biased (due to the limited sample size) and, consequently, also the scientific content of the research may be affected. Since Authors have explained that they have more samples and data from other 40 dogs, it would be more intresting to group together Golden retrievers from this manuscript (n: 10) with those that still need to be evaluated (n: 40) to obtain a final sample size of 50 dogs. Nevertheless, if Authors and Editors have a different opinion and still prefer to proceed with the publication of this pilot study, I have some additional comments that I hope could be useful for the manuscript.
We would really like to thank the Reviewer 1 for all his/her suggestions and for giving us the opportunity to decide whether to publish these preliminary data or add them to the data collected in recent months. Just today we added more samples, reaching 53 Golden Retrievers, but the taurine concentration data will probably arrive in a couple of months.
If the Editor also agrees and if the second revision is considered suitable, we prefer to start publishing the data of this preliminary pilot study.
In any case the revision of Reviewer 1 has been of great help to better focus the aim and give a greater overview of what is the literature on this breed. Thank you.
Title
I suggest to introduce the term “preliminary data”. as Authors have said that they likely will publish further data on this topic from a lager sample size. Moreover, I would suggest to introduce the numbers of dogs from each breed also in the title, so that readers can easily understand that data from this study should be interpreted cautiously, as it is a polit study providing preliminary data from a small study population.
According, the final version of the tile should be: “Preliminary data on echocardiographic evaluation and serum taurine concentration in healthy dogs of two breeds (10 Golden Retrievers and 12 German Shorthaired Pointers) with different predisposition to nutritional dilated cardiomyopathy: a pilot study”.
Thank you very much for the suggestion, we completely agree, and we have modified the title.
Simple summary
-Line 19: What does it mean “reduction of the cardiomyocytes hypertrophy”? Does it mean that a low VitD concentration may induce atrophy of cardiomyocytes? Please, explain it better.
Thank you for the comment. Yes, it has been demonstrated that vitamin D slowed cardiomyocyte proliferation, promoted cell cycle arrest and decreased expression of pro-fibrotic factors during TGF-beta induced fibroblastic differentiation. Furthermore, vitamin D induces an improvement of the thickness of the left ventricular wall and a significant improvement in left ventricular systolic function. We have rephrased the sentence in: “Furthermore, in human medicine it has been demonstrated that low vitamin D concentration may induce a thinning of cardiomyocytes and the onset of congestive heart failure”.
-Line 21: “Because taurine is involved in the digestion and absorption of fat and liposoluble vitamins”. Is VitD included among vitamins regulated by taurine? If yes, please change this sentence to: Because taurine is involved in the digestion and absorption of fat and liposoluble vitamins, including vitamin D, ….”
Thank you, yes, vitamin D is a liposoluble vitamin regulated by taurine. We have modified the sentence as you suggested.
-Lines 24-25: “Our results showed that GRs presented lower levels of serum taurine and more altered echocardiographic variables than GSPs”. Given the limits of the study (first of all, the small sample size), I suggest to rewrite this sentence as: “The preliminary findings from this small pilot study showed that GRs may present lower levels of serum taurine and more altered echocardiographic variables than GSPs”).
Thank you very much, we have modified the conclusion of the simple summary.
Abstract
-Lines 30-31: “Because taurine is involved in the digestion and absorption of fat and liposoluble vitamins, the aims of… Also in this sentence I would add the terms, “including vitamin D” (if this is scientifically correct), so that the final result would be: “Because taurine is involved in the digestion and absorption of fat and liposoluble vitamins, including vitamin D, the aims of…”
Introduction
Thank you, we modified as suggested and as in the simple summary.
-Lines 50-56: among the list of differential diagnosis for a DCM-like phenotype, it should be included also ischemic heart disease. Moreover, I suggest to introduce here the following additional references from manuscripts and congress abstract:
-manuscripts: Schneider SM, Coleman AE, Guo LJ, Tou S, Keene BW, Kornegay JN. Suspected acute myocardial infarction in a dystrophin-deficient dog. Neuromuscul Disord 2016;26:361-6. // Romito G, Cipone M. Transient deep and giant negative T waves in dogs with myocardial injury. J Vet Cardiol . 2021 Aug;36:131-140. // Driehuys S, Van Winkle TJ, Sammarco CD, Drobatz KJ. Myocardial infarction in dogs and cats: 37 cases (1985-1994). J Am Vet Med Assoc 1998;213:1444-8. // Falk T, Jo ̈nsson L. Ischaemic heart disease in the dog: a review of 65 cases. J Small Anim Pract 2000;41:97-103.
-congress abstarcts: Lekane M, Connolly D, Smets P, Borgeat K, Casamian-Sorrosal D, Boswood A, Luis Fuentes V, Gommeren K, Merveille AC. Clinical, ECG and echocardiographic findings in a canine case series of presumptive myocardial infarction. J Vet Intern Med 2020;34:403-4.
Thank you, we have added these references and modified all the reference’s numeration.
-Line 69: typing error “[,11,12]”.
Thank you very much, we have modified.
-Line 82: typing error “bredds”
Thank you very much, we have modified.
-Lines 110-111: “This would increase their intraluminal concentrations with possible improvement in absorption of fat and liposoluble vitamins [21]”. A greater absorption of vitamin D due to the positive role of taurine has been demonstrated in humans? If yes, change the sentence in this way: “This would increase their intraluminal concentrations with possible improvement in absorption of fat and liposoluble vitamins, including vitamin D [21]”.
Thank you, exactly, we have modified the sentence as suggested.
Moreover, please say if a relationship between VitD and taurine has been studied also in dogs and if similar results have been found in this species (because not always what is found in humans is identical in dogs).
Thank you very much. We have speculated on this topic.
-Line 111: add the word “pilot” so that the final version of the sentence become: “The aim of this pilot study…”. Please, add the word “pilot” in all other part of the manuscript (e.g., line 119), so that it becomes really clear to readers the type of study.
Thank you, we have added the word “pilot” as suggested. Furthermore, we added the term “pilot study” also in the key words’ list.
-Lines 134-135: Please change this sentence in this way: “Furthermore, to estimate serum taurine and vitamin D3, concentration venous blood sample was collected in fastened dogs (12-8h).”
Thank you, we have modified as you suggested.
-Line 155 and 158: there are a lot of a acronyms that appear here for the first time. It is essential to provide initially the extended explanation of the abbreviation (e.g., EDVI = end-diastolic left ventricular volume indexed). Then, it is important to check before submission how many times each abbreviation is used, in order to maintain only those used sufficient times according to rules of the Journal. Lastly, I am not sure that is allowed to start a sentence with an abbreviation.
Thank you for this comment. We have checked the acronyms, that are all explained in the introduction section. To the author knowledge, Animals MDPI doesn’t report specific rules for the number of acronyms citation, so we have not deleted the acronyms reported only one or two times in the main text. Furthermore, we have deleted the acronyms used at the beginning of a sentence and we have modified with the entire name of the measurement.
Results
-Section 3.1. named “Golden Retriever”: all this section has been written using italics.
Thank you. We have modified and used the italics inky for the title of the section 3.1. Thank you.
Discussion
-Line 264-265: “In our study, this aspect is confirmed by the finding of a lower serum taurine concentration in the GRs, …”. I don’t agree with the use of the term “is confirmed”, as a small study on 10 Golden retrievers cannot confirm a breed predisposition. Therefore, I suggest to rewrite this sentence as: “Our results appear overall in line with previous literature in light of the finding of a lower serum taurine concentration in the GRs, ...”.
Thank you, we have modified the sentence.
-lines 269.273: at the end of this long sentence I suggest to add “…, although further studies including more dogs are needed to conclusively confirm this”.
Thank you for this suggestion. We have added this speculation.
-Lines 285-286: “The results suggests that the serum and plasma taurine concentration does not differ…”. I suggest to introduce the word “may” to make less strong the statement, as it is difficult to obtain conclusive results given the small sample size. Moreover, there is a typing error. Accordingly, the final version of the sentence would be: “The results suggest that the serum and plasma taurine concentration may not differ…”.
Thank you, we have modified the sentence.
-Lines 328-331: “The higher values of the parameters suggestive of ventricular dilation and lower values of those predictive of systolic dysfunction, associated to a low levels of serum taurine, supports the hypothesis of a possible greater predisposition of the GRs to develop DCM.” I don’t completely agree with this statement as the small sample size of the study does not allow to make definitive conclusions at regards. Indeed, it could be also possible that the parameters suggesting a mildly larger dimensions of the LV and a mildly “more hypokinetic” LV in GR compared to GSP, may be not related to a disease state (i.e., they may be not due to a true hypokinetic cardiomyopathy related to taurine) but they may be simply due to physiological differences concerning normal echocardiographic parameters in the two breeds. The only way to know the answer would be to make a study on ≥120 healthy GR and ≥120 GPS to create statistically reliable normality ranges for standard echocardiographic parameters in these two breeds, and then to compare the reanalyze the results of this study. Therefore, all this should be discussed and clearly explained to readers.
Thank you for this interesting comment. We have modified the sentence as follow: “The higher values of the parameters suggestive of ventricular dilation and lower values of those predictive of systolic dysfunction, associated to a low levels of serum taurine, supports the hypothesis of a possible greater predisposition of the GRs to develop DCM or simply suggests a peculiar breed characteristic, not necessarily related to lower plasma concentration of taurine. This may or may not be confirmed by a study that compares the data of the two breeds and that includes more than one hundred dogs each, fed the same identical diet. At present, therefore, only assumptions can be made.
-lines 338-339: “The main limit of this study was the small number of subjects included.” I suggest to expand this section in this way: “The main limit of this study was the small number of subjects included, which inevitably weakened our statistical analysis and results. Accordingly, findings from this small pilot study should not be read as a guide for interpretation of taurine concentration as well echocardiographic parameters in these two breeds. In contrast, they should be simply interpreted as preliminary data aimed at paving the way for further and more robust investigations in this topic.”
Thank you very much, we decided to follow your suggestion and we added this speculation in the manuscript.
-Additional limitation: an additional limitation that should be included is that Author have selected two breeds for which there are not well-established echocardiographic normality ranges based on statistically reliable sample sizes. This inevitably makes it more difficult to correctly interpret the echocardiographic findings from this study and makes it easier to interpret as abnormal something that maybe could be normal for that given breed. This should be explained to readers.
Thank you very much for this comment. We speculated about this limitation adding a sentence in the text: “Furthermore, it must be highlighted that literature doesn’t report well-established echocardiographic normality ranges based on statistically reliable sample sizes for both GRs and GSPs. For this reason, it is difficult to correctly interpret the echocardiographic findings from this pilot study and makes it easier to interpret as abnormal something that maybe could be normal for a given breed”.
Conclusions
-lines 361-362: “These echocardiographic alterations/border line values could be considered as a predisposition of this breed to develop a dilatative/hypokinetic myocardial disease”. This sentence should be deleted.
Thank you very much for this suggestion. We have deleted this sentence.

Reviewer 2 Report
Congratulations, in my opinion the manuscript is ready to publish in Animals because the recommendations indicated previously have been included.
Author Response
The authors would like to thank Reviewer 2 for his/her decision and for helping us to improve our study. Thank youu very much.